# HiRes-LLaVA: Restoring Fragmentation Input in High-Resolution Large Vision-Language Models

## Abstract

High-resolution image inputs allow Large Vision-Language Models (LVLMs) to capture finer visual details, improving comprehension. However, the increased training and computational costs associated with such inputs pose significant challenges. A common approach to mitigate these costs involves slicing the input into uniform patches using sliding windows, each aligned with the vision encoder's input size. While efficient, this method fragments the input, disrupting the continuity of contextual, which negatively impacts cross-patch perception tasks. To address these limitations, we propose **HiRes-LLaVA**, a novel framework designed to efficiently process high-resolution inputs of any size without altering the original contextual and geometric information. HiRes-LLaVA introduces two key components: (i) a SliceRestore adapter (SRA) that reconstructs sliced patches into their original form, enabling efficient extraction of both global and local features through down-up-sampling and convolutional layers, and (ii) a Self-Mining Sampler (SMS) that compresses vision tokens based on internal relationships, preserving original context and positional information while reducing training overhead. To assess the ability of handling context fragmentation, we construct a new benchmark, EntityGrid-QA, consisting of edge-related tasks. Extensive experiments demonstrate the superiority of HiRes-LLaVA on both existing public benchmarks and EntityGrid-QA. For example, with SRA, our method achieves a performance improvement of $\sim 9\%$ over state-of-the-art LVLMs in addressing fragmentation issues. Additionally, our SMS outperforms other visual token downsamplers, while offering comparable efficiency.

## 1 Introduction

Recent progress in Large Vision-Language Models (LVLMs) (Alayrac et al., 2022; Li et al., 2023c;b;d; Liu et al., 2023f; Zhu et al., 2023) has significantly enhanced capabilities in vision-language tasks, fostering improved understanding, reasoning, and interaction. Early LVLMs (Li et al., 2023b; Zhu et al., 2023; Liu et al., 2023d) processed images at low resolutions, typically $224 \times 224$, which hindering their ability to capture detailed visual information. This limitation often results in inaccurate recognition of objects and their contextual relationships within images (Ding et al., 2023; Li et al., 2023e).

Enhancing the high-resolution capabilities of LVLMs presents substantial challenges, *i.e.*, training visual encoders to handle high-resolution inputs requires significant computational resources as well as struggling with handling arbitrary image sizes (Bai et al., 2023a; Chen et al., 2023c). Recent advances have introduced resource-efficient methods to improve the input resolution of LVLMs. One effective strategy involves using a sliding window technique (Li et al., 2023e; Xu et al., 2024a; Liu et al., 2024b) to segment high-resolution images into smaller patches. These patches are then processed by a visual encoder that has been trained on fixed-size lower-resolution inputs, maintaining computational efficiency while enhancing detail capture.

Although effective, this slicing approach leads to the fragmentation of the original input, resulting in a disruption of context. As illustrated in Fig.1, slicing the entire image can alter the original context, especially when an object is located at the edge of two slices. This slicing strategy makes

(a) Fragment input construction    (b) Evaluation for nine positions    (c) Visualization of accuracy

Figure 1: **Illustration of the fragmentation issue. (a)** We construct nine image inputs with objects placed in nine different positions. Four of these positions, *i.e.*, (2,4,6,8) are located at the edges of two slices, resulting in fragmentation issues. **(b)** We input these nine images along with corresponding questions into the LVLMs to evaluate the accuracy of object recognition at different positions. Note that the green circles with numbers are for illustration purposes only and are not utilized by the LVLMs. **(c)** The visualization of accuracy at various positions demonstrates that our method outperforms both slicing-based and non-slicing methods across all positions.

it more challenging for the model to identify the fragmented objects and text, thereby hindering the model's overall understanding of the image and impeding its ability to perform more complex cognitive tasks. Furthermore, existing approaches (Xu et al., 2024a; Liu et al., 2024b) generally use a sampler, such as Q-Former (Li et al., 2023c), to reduce the long context caused by high-resolution input. However, this plain Q-Former like sampler utilizes a fixed number of queries to compress and capture visual features through a cross-attention mechanism, suffering from problems, *e.g.*, lacking position information and high training overhead Yao et al. (2024).

In this paper, we propose HiRes-LLaVA, an efficient approach to integrating high-resolution data into LVLMs without disrupting the original context. As illustrated in Fig.1 (c), our method maintains consistent accuracy even when objects are positioned across different slices. HiRes-LLaVA utilizes a new SliceRestore Adapter to combine sliced low-resolution patch features into a high-resolution feature map, preserving the image's complete context. This map is processed through dual parallel fusion modules to capture both global and local information. The enhanced high-resolution map is then segmented back into small patches. The SliceRestore Adapter is a lightweight module that can be seamlessly integrated into any attention layer of the low-resolution vision encoder, enabling efficient fine-tuning without altering pre-trained parameters. Furthermore, we introduce a self-mining sampler that uses average pooled sliced patches as queries. Unlike fixed learnable query-based methods, our self-mining sampler preserves the original context and positional information while optimizing efficiently.

To evaluate our proposed method, we tested it on nine widely-used public benchmarks and also introduced a new benchmark, EntityGrid-QA, specifically designed to measure how well VLMs handle context fragmentation caused by slicing approaches. Our comprehensive experiments show that HiRes-LLaVA not only performs better than current models on these public benchmarks but also significantly surpasses SOTA LVLMs over $\sim 9\%$ on the EntityGrid-QA benchmark. Additionally, our SMS outperforms other visual token downsampling methods, all while maintaining similar efficiency.

## 2 RELATED WORKS

**Large Vision-Language Model.** Leveraging pre-trained Large Language Models (LLMs) like LLaMA (Touvron et al., 2023) and Vicuna (Chiang et al., 2023), Large Vision-Language Models (LVLMs) have achieved significant advancements in areas such as image/video understanding (Li et al., 2022; 2023c; Zhu et al., 2023; Alayrac et al., 2022; Chen et al., 2023a; Zhang et al., 2023a; Li et al., 2023d), medical analysis (Li et al., 2023b), and autonomous driving (Ding et al., 2023; Xu et al., 2023). These models utilize vision encoders trained via contrastive learning (Dosovitskiy et al., 2020; Radford et al., 2021) to align visual features with language. Visual embeddings are then adapted to match LLM dimensionality through visual projectors, which can be (i) learned queries, like the perceiver resampler (Alayrac et al., 2022) or Q-Former (Li et al., 2023c; Zhu et al., 2023), using fixed queries for cross-attention, or (ii) MLP modules, as seen in the LLaVA series (Liu et al.,

2023f). Recent efforts have aimed to enhance visual representation by combining features from DINO-V2 (Oquab et al., 2023) and SAM (Kirillov et al., 2023) with CLIP's Vision Transformers (ViT) (Ranzinger et al., 2023; Lin et al.). However, the reliance on CLIP-ViT, which requires fixed-resolution images (e.g., $336 \times 336$), limits the capability to handle higher resolutions and varying aspect ratios, thereby hindering performance in fine-grained tasks.

**High Resolution Large Vision-Language Model.** To discern fine-grained visual details from high-resolution inputs, an intuitive approach is to split images into patches and project them using linear layers, treating these as a sequence for input into Large Vision-Language Models (LVLMs) (Bavishi et al., 2023; Li et al., 2023a). While this eliminates the need for an image encoder, it often results in insufficient visual representation, leading to increased training costs and suboptimal performance. Alternatively, Up-Resize methods such as Qwen-VL (Bai et al., 2023a) adapt the positional embeddings of ViT from $224 \times 224$ to $448 \times 448$ and include an additional training phase to fine-tune the ViT. However, this adaptation may alter the original visual position encoding from CLIP-ViT (Radford et al., 2021), potentially degrading visual representation. Dual-branch approaches introduce a high-resolution branch with lightweight convolutional networks to manage high-resolution inputs but require additional training data and parameters (Hong et al., 2023; Ding et al., 2023; Luo et al., 2024; Li et al., 2024a). Slicing-based methods offer a compromise by using slicing windows to divide the high-resolution image into patches that match the input size of a pre-trained vision encoder, maintaining efficiency in parameter use and training data while still achieving competitive performance (Li et al., 2023e; Xu et al., 2024a). However, they suffer from "Context Fragmentation", where the continuity of contextual information across patches is damaged, impacting tasks that require cross-patch context. In this paper, we propose HiRes-LLaVA, a novel technique designed to seamlessly integrate global-local high-resolution details into LVLMs without disrupting the original context, effectively addressing the issue of Context Fragmentation.

## 3 METHOD

In this section, we first present the overall framework of HiRes-LLaVA in Section 3.1. The two innovative components, namely SliceRestore adapter and self-mining sampler are detailed in Section 3.2 and Section 3.3 respectively. To further evaluate the ability of VLMs to address the context fragmentation issue, a new benchmark named EntityGrid-QA is proposed in Section 3.4.

### 3.1 OVERALL FRAMEWORK

The overall framework of HiRes-LLaVA is shown in Fig. 2. First, the original image is resized and padded to a low resolution (typically $224 \times 224$) and processed by the pre-trained vision encoder, producing global features. To capture fine-grained details, the high-resolution image is split into slices by a dynamic slicing strategy. Detailedly, we set a maximum slice count $M$, allowing an image to automatically select an optimal bounding box by calculating the necessary $m$ rows and $n$ columns based on the base resolution:

$$m = \left\lceil \frac{H}{r} \right\rceil, n = \left\lceil \frac{W}{r} \right\rceil.$$

where $r$ is the base resolution in pretrained vision encoder. This slicing approach adapts to the image's original aspect ratio, only quadrupling the number of slices by scaling $2\times$ of $m$ and $n$ if "$4 * m * n$" does not exceed $M$, ensuring detailed preservation without overwhelming the model. Afterwards, these slices are processed by a shared vision encoder with the proposed SliceRestore adapter, yielding slice features, followed by a shared self-mining sampler to reduce token length, resulting in compressed features. As a result, our visual input to the language model includes a low-resolution overview and multiple high-resolution slices, which also differentiated by three types of separators to maintain clarity in (1) between the low-resolution image and high-resolution slices, (2) between high resolutions slices and (3) the end of each slice row.

### 3.2 SLICERESTORE ADAPTER

As depicted in Fig. 2 (a), the SliceRestore adapter is integrated into the self-attention layer of vision transformer. We denote the slice features in the $l$-th layer of ViT as $\{\mathbf{P}_i\}_{i=1}^N$ with $\mathbf{P}_i \in \mathbb{R}^{L \times D}$, where $N$ is the number of slices, $L = H \times W$ is the token length, and $D$ is the feature dimension. Each slice feature is processed individually by the self-attention layer, *Self-Attn*($\mathbf{P}_i$), which lead to a loss of global information in fragmented context. (see Fig. 1 (a)). Although low-resolution inputs contain

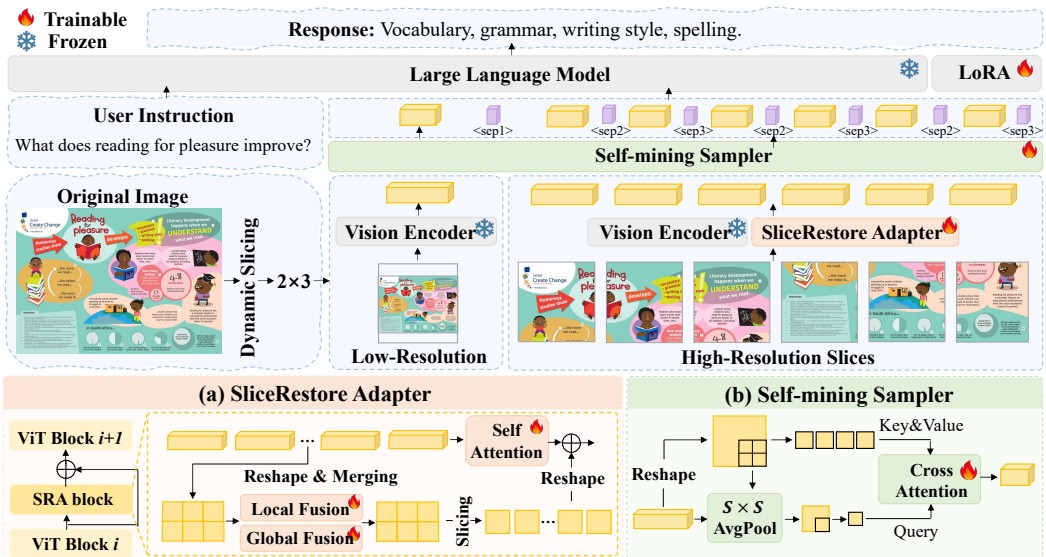

Figure 2: **Overall framework of HiRes-LLaVA.** The vision encoding consists of two branches: one for low-resolution images processed by the pre-trained vision encoder to extract global features, and another dividing high-resolution images into multiple slices to capture fine-grained details. **(a) SliceRestore Adapter** aims to address the Context Fragmentation issue, it restores sliced features into a whole feature by capturing both local and global information, then splits the whole feature back into slices. **(b) Self-Mining Sampler** compresses vision token numbers to reduce computation and memory costs by using downsampled features as queries and the original features as keys and values. Both low-resolution image input and each high-resolution slice are compressed by the same self-mining sampler.

the overall information, when it comes to real-world scenes, small objects in slices are still difficult to perceive. A naive approach would be concatenating slice features for self-attention, but this incurs quadratic computation costs.

In this paper, we propose the SliceRestore Adapter (SRA) to efficiently capture complete information from high-resolution inputs. This is formulated as:

$$\{\hat{\mathbf{P}}_i\}_{i=1}^N = \{\mathbf{P}_i\}_{i=1}^N + \{\textit{Self-Attn}(\mathbf{P}_i)\}_{i=1}^N + \{\overline{\mathbf{P}}_i^l\}_{i=1}^N, \tag{1}$$

where:

$$\{\overline{\mathbf{P}}_i^l\}_{i=1}^N = SRA(\{\mathbf{P}_i\}_{i=1}^N), \tag{2}$$

The SliceRestore adapter has three main steps to restore complete semantics from slice features:

**1. Merging**: Each slice feature $\mathbf{P}_i$ is first reshaped into $\mathbf{H}_i \in \mathbb{R}^{H \times W \times D}$. These reshaped slice features, $\{\mathbf{H}_i\}_{i=1}^N$, are then recover the original spatial structure and merged to form the original input's features $\mathbf{F} \in \mathbb{R}^{(m*W) \times (n*H) \times D}$. $m$ and $n$ indicate the number of slices' rows and columns, respectively. $N$ is equal to $m * n$.

**2. Capturing**: We propose two fusion modules for extracting both local and global information from $\mathbf{F}$. The local fusion module focuses on transferring edge details among slices, facilitating a nuanced exchange of local information. On the other hand, the global fusion module is leveraged to capture broader contextual cues. To achieve this, The local fusion module uses a single layer depth-wise convolution with $3 \times 3$ kernel to efficiently capture local details and retain image-related biases. The global fusion module employs self-attention to capture the global context. Given the quadratic computation cost of self-attention, we first downsample $\mathbf{F}^l$ to create an overview of the image in a smaller size, i.e., the same size of the low-resolution image and feed it to a self-attention block and then upsample back to the original size, by simpling using an interpolation. The enhanced whole feature $\overline{\mathbf{F}}$ is obtained by element-wise addition of the outputs from the local and global fusion

Figure 3: **Construction process of EntityGrid-QA benchmark.** There are three steps: **(a) Entity Sampling.** Select one or two entities from the pre-defined entity set; **(b) Image Generation.** Put the selected entities in one position sampled from the nine pre-defined positions of the blank image, we can obtain the generated images. Note that the dash and solid lines in (b) are for illustration purposes only, and not presented to models. **(c) QA pairs Generation.** Based on the generated images, entity category and positions, we can automatically generate the question-answer pairs (QAs).

modules:

$$\overline{\mathbf{F}} = \underbrace{\textit{Depth-Wise Conv}(\mathbf{F})}_{\text{local fusion}} + \underbrace{\text{Up}(\textit{Self-Attn}(\text{Down}(\mathbf{F})))}_{\text{global fusion}}. \tag{3}$$

**3. Slicing**: Finally, the enhanced whole feature $\overline{\mathbf{F}}$ is sliced back into the original slice format, resulting in $\{\overline{\mathbf{P}}_i\}_{i=1}^N$, where $\overline{\mathbf{P}}_i \in \mathbb{R}^{L \times D}$.

This process allows model to capture the complete semantics from high-resolution inputs while maintaining computational efficiency.

### 3.3 SELF-MINING SAMPLER

High-resolution images necessitate processing significantly more visual tokens, contributing to a substantial part of the computational load. Existing solutions, such as Q-Former (Li et al., 2023c), utilize a fixed number of queries to compress and capture visual features through a cross-attention mechanism. While this method effectively captures visual information regardless of image resolution in a computationally affordable manner, it suffers from several limitations: **(i) Lacking Positional Information.** Learned queries lose positional information, degrading performance in tasks requiring spatial relationships and precise localization. **(ii) High Training Overhead.** Training Q-Former-like resamplers requires more data and longer training times to convert visual features into learnable queries, posing challenges in data-scarce domains.

To address the issue, we propose the self-mining sampler, as shown in Fig. 2 (b). The key idea of the self-mining sampler is to better initialize the query and narrow the receptive field that per query needs to compress. Specifically, we reshape the one-dimensional output vision tokens of the vision encoder (e.g., CLIP-ViT), $\mathbf{P} \in \mathbb{R}^{L \times D}$, into a two-dimensional form, $H \times W \times D$, where $L = H \times W$. After applying average-pooling with kernel size $S \times S$, we obtain $\mathbf{P}^c \in \mathbb{R}^{H_2 \times W_2 \times D}$, where $W_2 < W$ and $H_2 < H$. Next, we compute the final compressed tokens using the cross-attention mechanism, *Cross-Attn*$(\mathbf{P}^c, \mathbf{P})$, with $\mathbf{P}^c$ as the query and $\mathbf{P}$ as the key and values. Compared with fixed learnable query-based methods, our self-mining sampler compresses the vision tokens based on themselves, preserving the original context and positional information while reducing training overhead.

### 3.4 ENTITYGRID-QA BENCHMARK

Existing benchmarks, particularly document-related datasets, can evaluate the fine-grained understanding of LVLMs. However, these benchmarks are inadequate for assessing the ability to handle fragmented inputs, as filtering slicing-related questions is time-consuming and labor-intensive. Therefore, we introduce a new benchmark named EntityGrid-QA, which is fully synthesized but still challenging for frontier models, to better assess LVLMs' ability to handle fragmentation.

**Construction Process.** As shown in Fig. 3, the construction process of EntityGrid-QA consists of three main steps: Entity Sampling, Image Generation, and QA Pairs Generation. Examples of our benchmark are provided in the Appendix. Each step is detailed as follows:

**(a) Entity Sampling**. We first construct an entity set that includes various types such as English Words (*e.g.*, "apple"), Number (*e.g.*, "0.596"), Object (*e.g.*, a teddy bear) and Icon (*e.g.*, "tomato") as shown in Fig. 3 (a). Then, we select several entities from a predefined entity set, which can be denoted as $\mathcal{E} = \{e_i\}_{i=1}^{M}$, where $e_i$ is the $i$-th entity and $M$ is the number of selected entities.

**(b) Image Generation**. The selected entities $\mathcal{E}$ are positioned in nine predefined positions (labeled 1 to 9) within a blank image $I$ using a 3x3 grid layout, as shown in Fig. 3 (b). The resolution of the blank image is set to $2R$, where $R$ is the base resolution for existing LVLMs, *e.g.*, $224 \times 224$. In this way, each $I$ would be divided into four slices during inference, and the each slice would match the input size of well-pretrained vision encoder, without the requirement of additional operations, *e.g.*, resize and padding. Note that our HiRes-LLaVA can process any number of slices, but some existing LVLMs, *i.e.*, LLaVA-Next (Liu et al., 2023d) can only receive four slices as input. Hence, for a fair comparison, we only generate the images with a fixed resolution $2R \times 2R$. For each entity $e_i$, we generate $P$ images that iterate over all predefined positions, *i.e.*, 9 positions as shown in Fig. 3.

**(c) QA Pairs Generation**. We mainly focus on evaluating the model's fine-granite recognition ability on the area of the slice boundary and center of the slices. For each type of entity, we apply a specific question prompt, *e.g.*, `"What is the object in the picture?"`. As shown in Fig. 3 (c), we formulate the question-answer pairs as the multiple choice problem. Based on the selected entity $\mathcal{E}$ and the question $Q$, we apply the entity-specific augmentation to automatically generate the other three choices for the question. For example, given a number, the optional augmentations can be add, delete or shift the decimal point, or alter one of the digit of the number. The ground truth option letter the answer, can be obtained by comparing the choices with the selected entity. Note that for the triplets of image-question-answer of the same entity, it only varies in the position of the generated images $I$ while maintaining the same question, order of choices and ground truth answer which is perfectly assess the model.

After the construction, we create a training set of Entity-QA with 7k images covering 4 type of entities and a testing set with 3.6K images and 20 entities for each type. Note that the entities are non-overlapped between the training set and testing set. The examples of the benchmark can be found in the Appendix.

**Evaluation Metric.** To evaluate the ability to handle the fragmentation, we introduce a new metric that measures the precision discrepancies between entities located at the edge positions ($\mathcal{P}_{\text{edge}} = \{2, 4, 5, 6, 8\}$) and other locations ($\mathcal{P}_{\text{center}} = \{1, 3, 7, 9\}$), defined as:

$$\text{Discrepancy}_1 = \frac{\sum_{p \in \mathcal{P}_{\text{edge}}} A_p / |\mathcal{P}_{\text{edge}}|}{\sum_{p \in \mathcal{P}_{\text{center}}} A_p / |\mathcal{P}_{\text{center}}|}, \tag{4}$$

$$\text{Discrepancy}_2 = \frac{\sum_{p \in \mathcal{P}_{\text{edge}}} A_p / |\mathcal{P}_{\text{edge}}| - \sum_{p \in \mathcal{P}_{\text{center}}} A_p / |\mathcal{P}_{\text{center}}|}{\sum_{p \in \mathcal{P}_{\text{center}}} A_p / |\mathcal{P}_{\text{center}}|}, \tag{5}$$

where $A_p$ is the average accuracy of three tasks when entities located at the position $p$, $|\cdot|$ is the number of elements in the set.

# 4 EXPERIMENT

## 4.1 IMPLEMENTATION DETAILS

We utilize the CLIP-ViT-L/14-224px (Radford et al., 2021) and InternViT-300M-448px as the vision encoders, and Vicuna-v1.5-7B (Chiang et al., 2023) and LLaMA-3.1-8B (Dubey et al., 2024) as LLM. We adopt a three-stage training approach, including an alignment stage, a capability enhancement stage and the instruction tuning stage. During the alignment stage, only the self-mining sampler is trainable. The learning rate is 1e-3. In the capability enhancement stage, both full model including the vit, sampler and LLM is unfreezed. The learning rate is 2e-5 for LLM and sampler and 2e-6 for ViT. In the instruction tuning stage, ViT is freezed and the SliceRestore adapter is loaded with the LR of 2e-4. The learning rate of self-mining sampler and LLM is 2e-5. Four SliceRestore adapter are applied in the last four blocks of the vision encoder. All stages use the batch size of 256. We adopt AdamW (Loshchilov & Hutter, 2017) as the optimizer with $\beta_1 = 0.9$ and $\beta_2 = 0.95$ to stabilize the training in the capability enhancement stage and the instruction tuning stage. In all stages, the learning rates are warmed up for the first 0.03 epochs and then adjusted by a cosine scheduler in the

Table 1: **Quantitative results on 9 popular benchmarks.** 'MaxRes' means the maximum resolution supported. 'Doc', 'Science' and 'Comprehensive' indicate the document-related VQA, Science VQA and comprehensive benchmarks.

| Method | LLM | MaxRes | Doc | | | | Science | | Comprehensive | | |
|--------|-----|--------|----------|---------|-------|---------|------|------|------|-----|--------|
| | | | VQA-text | ChartQA | DocVQA | InfoVQA | SQAI | ai2d | MME | MMB | MM-Vet |
| *General LVLMs (normal resolution)* | | | | | | | | | | | |
| Qwen-VL-Chat | Qwen-7B | 448×448 | 61.5 | 66.3 | 62.6 | - | 68.2 | 57.7 | - | 60.6 | - |
| LLaVA-1.5 | Vicuna-1.5-13B | 336x336 | 61.3 | 18.2 | - | - | 71.6 | 59.5 | 1826 | 67.8 | 36.3 |
| LLaVA-MORE | LLaMA3.1-8B | 384x384 | 62.1 | - | - | - | 77.5 | 63.6 | 1846 | 73.1 | - |
| mPLUG-Owl3 | Qwen1.5-7B | 384x384 | 69.0 | - | - | - | - | 73.4 | - | 77.6 | 40.1 |
| *Document LVLMs* | | | | | | | | | | | |
| DocPedia | Vicuna | 2560×2560 | 60.2 | 46.9 | 47.1 | 15.2 | - | - | - | - | - |
| UReader | Vicuna | 896×1120 | 57.6 | 59.3 | 65.4 | 42.2 | - | - | - | - | - |
| TextMonkey+ | QWen-7B | 896x896 | 64.3 | 59.9 | 66.7 | 28.6 | - | - | - | - | - |
| mPLUG-DocOwl2 | Qwen2-7B | 1512x2016 | 66.7 | 70.0 | 80.7 | 46.4 | - | - | - | - | - |
| *General LVLMs (higher resolution)* | | | | | | | | | | | |
| Monkey | Qwen-7B | 896x896 | 67.6 | - | 66.5 | 36.1 | - | - | - | - | - |
| LLaVA-NeXT-8B | LLama3-8b-Ins | 672x672 | 64.6 | 69.5 | 72.6 | - | - | 71.6 | 1603/- | 72.1 | 41.7 |
| LLaVA-NeXT-13B | Vicuna-13B | 672x672 | 67.1 | 62.2 | 70.9 | - | 73.6 | 70.0 | 1901 | 70.0 | 48.4 |
| LLaVA-UHD | Vicuna-13B | 672×1008 | 67.7 | - | - | - | 72.0 | - | 1535/- | 68.0 | - |
| Mini-Gemini-HD | Llama3-8b-Ins | 672x672 | 70.2 | 59.1 | 74.6 | - | 75.1 | 73.5 | 1606/- | 72.7 | - |
| Cambrian-1-8B | Llama-3-Ins-8B | 1024x1024 | 71.7 | 73.3 | 77.8 | - | 80.4 | 73.0 | 1547/- | **75.9** | - |
| Cambrian-1-13B | Vicuna-1.5-13B | 1024x1024 | 72.8 | 73.8 | 76.8 | - | 79.3 | 73.6 | 1610/- | 75.7 | - |
| **HiRes-LLaVA** | Llama-3.1-Ins-8B | 1344x1344 | **74.2** | **77.4** | **84.9** | **55.7** | **90.3** | **74.9** | **2213** | 75.7 | **53.5** |

remaining training. We don't apply any weight decay in the training. The maximum number of slices is 9 for InternViT and 16 for CLIP-ViT. Regarding the training data, we use the LLaVA-558k In the alignment stage, 1.8M caption and OCR data in the capability enhancement stage and 3M multi-tasks instruction data in the instruction tuning stage.

## 4.2 EXPERIMENTAL SETTING

We introduce experimental settings including the benchmarks and the compared LVLMs.

**Benchmarks.** We evaluate our models on four document-related VQA benchmarks, including VQA-text(Singh et al., 2019), ChartQA test set (Masry et al., 2022), DocVQA test set (Mathew et al., 2021), InfoVQA test set (Mathew et al., 2022), two general VQA benchmarks, including AI2D (Kembhavi et al., 2016), ScienceQA (Lu et al., 2022), and three comprehensive benchmarks, including MMBench (Liu et al., 2023g), MME (Fu et al., 2023) and MM-Vet (Yu et al., 2023).

**LVLMs.** We compare our model with SOTA LVLMs. (1) General baselines, *i.e.*, Qwen-VL (Bai et al., 2023a), LLaVA-1.5 (Liu et al., 2023d), mPLUG-Owl3 (Ye et al., 2024), Monkey (Li et al., 2023e), Mini-Gemini (Li et al., 2024b), LLaVA-UHD (Xu et al., 2024a), LLaVA-NeXT (Liu et al., 2024a) and Cambrian-1 (Tong et al., 2024), as representative general baselines. (2) Document LVLMs, *i.e.*, DocPedia (Feng et al., 2023), UReader (Ye et al., 2023), mPLUG-Docowl2 (Hu et al., 2024), TextMonkey (Liu et al., 2024b).

## 4.3 STATE-OF-THE-ART COMPARISON

**General Benchmarks.** Table 1 reports the performance comparison of our methods against state-of-the-art approaches on 11 benchmarks. Unexpectedly, our method utilizing LoRA fine-tuning (Hu et al., 2021) surpasses well-established LVLMs that require substantial data and extensive full fine-tuning, underscoring our model's efficiency and effectiveness. Notably, although both our model and Monkey (Li et al., 2023e) employ LoRA, Monkey is initialized from the pre-trained Qwen model (Bai et al., 2023b), while our model is trained from scratch, which further proves our model's efficiency. Furthermore, our method demonstrates competitive performance against specialized document-centric LVLMs such as TextMonkey (Liu et al., 2024b), proving its capability to manage document-related tasks effectively.

Figure 4 shows a visual comparison of results generated by LLaVA-Next (Liu et al., 2023d), Monkey (Li et al., 2023e), and our method, highlighting our superior performance, especially when the region of interest spans across slices. For example, the number $1.14$ in Fig. 4 (b) is split into two slices, causing Monkey to misrecognize it as $1.4$. Additionally, the slicing operation separates the

Table 2: **Comparison with the state-of-the-art methods on EntityGrid-QA.** '↓' indicates lower scores are better, while '↑' means higher scores are better. 'Accuracy$_{mean}$' and 'Accuracy$_{std}$', representing the mean and standard deviation of the average accuracy across three tasks. 'Accuracy$_{edge}$' and 'Accuracy$_{center}$' show the average accuracy for entities at $\mathcal{P}_{edge}$ and $\mathcal{P}_{center}$, respectively. Discrepancy$_1$ and Discrepancy$_2$ are calculated using Eq. 4 and Eq. 5. Note that IXC4KHD and HiRes-LLaVA are evaluated on 896x896 images and LLaVA-Next is evaluated on 672x672 images. The input resolution for LLaVA is 336px.

| Model | $Accuracy_{mean}\uparrow$ | $Accuracy_{std}\downarrow$ | $Accuracy_{edge}\uparrow$ | $Accuracy_{center}\uparrow$ | $Discrepancy_1\uparrow$ | $Discrepancy_2\downarrow$ |
|---|---|---|---|---|---|---|
| LLaVA | 53.33 | 0.19 | 52.0 | 55.00 | 94.50 | 5.45 |
| LLaVA-NeXT | 65.22 | 0.30 | 61.80 | 69.50 | 88.92 | 11.07 |
| IXC-4KHD | 63.78 | 0.53 | 58.00 | 71.00 | 81.69 | 18.31 |
| HiRes-LLaVA | 71.56 | 0.19 | 68.40 | 75.50 | 90.59 | 9.40 |

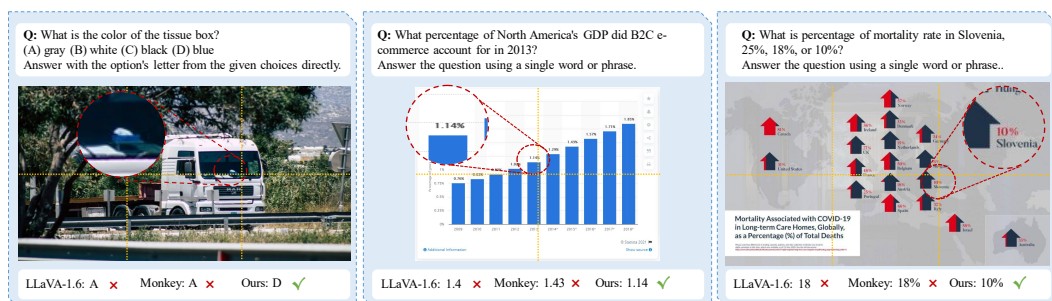

Figure 4: **The visualization comparison with the state-of-the-art methods.** Dash lines are only illustrated for the slice clarify, not presented to LVLMs.

year and percentage values into different slices, leading LLaVA-Next to incorrectly associate the 2017 percentage with 2014 due to the lack of global information. Our method, with the SliceRestore adaptercapturing complete global high-resolution information, correctly predicts the answers.

**EntityGrid-QA.** To evaluate the ability to address input fragmentation, we compare four SOTA slicing-based LVLMs with our HiRes-LLaVA and present the results in Table 2. According to the experimental results, we can observe two key findings: (i) Our method performs competitively on tasks with entities at $\mathcal{P}_{center}$. For instance, our method scores 71.56% on Accuracy$_{mean}$ and 75.50% on Accuracy$_{center}$, compared to the best prior SOTA scores of 63.78% and 71.00%. (ii) Our method significantly outperforms SOTAs in handling entities at $\mathcal{P}_{edge}$. For example, the previous SOTA, InternLM-Xcomposer-4KHD (Zhang et al., 2023b), shows a notable difference between Accuracy$_{edge}$ and Accuracy$_{center}$, with 58.0% vs. 71.0%. In contrast, our method achieves a smaller difference, with 68.4% on Accuracy$_{edge}$ and 75.5% on Accuracy$_{center}$. Additionally, the values of Discrepancy$_1$ and Discrepancy$_2$ further reflect the consistent performance of our method for both edge and center cases, surpassing existing SOTAs. In summary, our HiRes-LLaVA demonstrates superior ability to handle input fragmentation while maintaining competitive performance on center cases.

## 4.4 ABLATION STUDY

In this section, we conduct ablation studies to evaluate the effect of our proposed modules. In our ablation study, we conduct the experiments following LLaVA's setting on the LLaVA 1.2M data (Liu et al., 2023d) with additional 79K document-oriented data, which is essential to evaluate the high-resolution VLLMs, in the instruction tuning stage, i.e., DocVQA (Mathew et al., 2021), ChartQA (Masry et al., 2022) and InfoVQA (Mathew et al., 2022). Unless specified, we use LoRA (Hu et al., 2021) to efficiently finetune pretrained LLM, *i.e.*, Vicuna-1.5-7B and CLIP-ViT-Large-224px as the vision encoder with maximum 16 slices in our ablation.

**Effect of the proposed modules.** We ablate the two main components of our HiRes-LLaVA, specifically the SliceRestore adapter (SRA) and the self-mining sampler (SMS), as shown in Table 3. Our findings are as follows: Our SMS demonstrates superior performance compared to other samplers, notably outperforming Resampler (Bai et al., 2023b) by 6.9% on the average score across four benchmarks. Integrating the model with SRA leads to further improvements across these benchmarks.

Table 3: **The ablation study of different proposed modules.** Note that 'G', 'L', and 'G-L' indicate using the global fusion, the local fusion, and the combination of them respectively. All results are conducted with the maximum number of slices is 16 except the baseline model, LLaVA. The last row is the improvement over the baseline model.

| Components | | | Doc | | | | | Comprehensive | | |
|---|---|---|---|---|---|---|---|---|---|---|
| Downsampler | SRA | Separator | VQA-Text | ChartQA | DocQA | InfoVQA | Avg. | MMBench | MM-Vet | MME-P |
| **Baseline(LLaVA)** | | | 53.3 | 23.8 | 22.6 | 26.0 | 31.4 | 64.0 | - | 1424.7 |
| ConcatChannel | ✗ | ✗ | 60.3 | 54.4 | 54.8 | 34.3 | 50.9 | 60.8 | 30.2 | 1355.5 |
| Resampler | ✗ | ✗ | 58.8 | 49.8 | 42.8 | 32.6 | 46.0 | 59.6 | 26.6 | 1404.0 |
| C-Abstractor | ✗ | ✗ | 59.0 | 55.6 | 54.7 | 36.7 | 51.5 | 63.5 | 30.4 | 1393.5 |
| SMS | ✗ | ✗ | 60.0 | 56.2 | 58.0 | 37.4 | 52.9 | 63.3 | 31.1 | 1411.3 |
| SMS | G | ✗ | 60.9 | 56.2 | 57.2 | 38.2 | 53.1 | 65.5 | 30.6 | 1415.8 |
| SMS | G & L | ✗ | 61.5 | 56.9 | 57.6 | 38.4 | 53.6 | 64.9 | 33.8 | 1452.9 |
| SMS | G & L | ✓ | 61.8 | 58.8 | 59.7 | 41.4 | 55.4 | 65.5 | 33.8 | 1456.1 |
| improvement relative to the baseline | | | **+8.5** | **+35.0** | **+37.1** | **+15.4** | **+24.0** | **+1.5** | - | **+31.4** |

Table 4: **The ablation study of different vision encoder and large language models.** Note that CLIP-ViT-Large-224px uses 16 maximum slices and InternViT-300m-448px uses 9 slices.

| Components | | Doc | | | | | Comprehensive | |
|---|---|---|---|---|---|---|---|---|
| Vision Encoder | LLM | VQA-Text | ChartQA | DocQA | InfoVQA | Avg. | MMBench | MME-P |
| CLIP-ViT-Large-224px | Vicuna-1.5 | 61.8 | 58.8 | 59.7 | 41.4 | 55.4 | 65.5 | 1456.1 |
| CLIP-ViT-Large-224px | LLaMA3.1 | 60.5 | 58.6 | 67.2 | 47.2 | 58.4 | 68.1 | 1453.4 |
| InterViT300m-448px | LLaMA3.1 | 63.4 | 65.9 | 74.4 | 53.2 | 64.2 | 68.0 | 1459.1 |

Table 5: **The effect of different numbers of slices.** 'Max # Slices' indicates the maximum number of slices in the high-resolution images. 'Max # V Tokens' indicates the maximum number of visual tokens.

| | | Doc | | | | | Comprehensive | |
|---|---|---|---|---|---|---|---|---|
| Max #Slices | Max #Tokens | VQA-Text | ChartQA | DocQA | InfoVQA | Avg. | MMBench | MME-P |
| 4 | 320 | 56.2 | 42.5 | 37.0 | 28.8 | 41.1 | 65.1 | 1436.3 |
| 9 | 640 | 59.9 | 51.6 | 49.3 | 34.9 | 48.9 | 64.3 | 1450.0 |
| 16 | 1088 | 61.8 | 58.8 | 59.7 | 41.4 | 55.4 | 65.5 | 1456.1 |

Additionally, the introduction of learnable queries to isolate slice representations, referred to as Separator, results in a $1.8\%$ enhancement in the average score.

**Ablation study of kernel sizes in the self-mining sampler.** Here we conduct the ablation study of the self-mining sampler. In Table 6, we compare the performance of the average pooling with different kernel sizes, *i.e.*, $s \times s$ in Section 3.3. The results show that as the kernel size increases, *i.e.*, the fewer vision tokens, the performance would degrade, since the information loss.

**Ablation study of the number of high-resolution image slices.** As shown in Table. 5, the number of slices significantly affects the model's performance on the document-related benchmarks. Specifically, when increasing the number of slices from 4 to 16, the average performance improves by 14.3% on four document-related benchmarks. As for the comprehensive benchmarks, larger number of slices doesn't effect model's performance on MMBench too much and can bring a 19.8 improvement on MME-Perception. Although the trend of the performance illustrates that applying higher slices might bring more benefits, it will highly increase the computational cost during the training, i.e., 25 slices double the number of visual tokens of 16 slices. Balancing between the efficiency and the performance, We use 9 slices for the InternViT-300M in our main experiments.

**Ablation study of the selection of vision encoder and language model.** In Table 4, we evaluate the performance of different vision encoders and large language models on LVLM Benchmarks. Experimental results show that compared to Vicuna-1.5-7B, LLaMA3.1-8B-Instruct can signifi-cantly improve the model's performance on both document-related benchmarks and comprehensive benchmarks. Additionally, InternViT-300M-448px can maintain performance on comprehensive

Table 6: **Effect of different downsample kernel sizes in the self-mining sampler.** 'Downsample Kernel Size' is $S \times S$ defined in Section 3.3. 'Base Resolution' indicates the base resolution of the vision encoder. 'Max # V Tokens' indicates the maximum number of visual tokens, *i.e.*, $H_2 \times W_2$, as the maximum number of slices is 16.

| Base Resolution | Downsample Kernel Size | Max # V Tokens (Token/Slice) | Doc | | | | |
|---|---|---|---|---|---|---|---|
| | | | VQA-Text | ChartQA | DocVQA | InfoVQA | Avg. |
| 224 | $2 \times 2$ | 1088 (64) | 61.8 | 58.8 | 59.7 | 41.4 | 55.4 |
| 224 | $4 \times 4$ | 272 (16) | 59.6 | 53.9 | 46.3 | 33.0 | 48.2 |
| 224 | $8 \times 8$ | 68 (4) | 54.9 | 46.8 | 35.3 | 29.6 | 41.7 |
| 336 | $2 \times 2$ | 2448 (144) | 63.6 | 58.5 | 65.7 | 40.7 | 57.1 |
| 336 | $3 \times 3$ | 1088 (64) | 61.2 | 56.7 | 59.8 | 38.7 | 54.1 |
| 336 | $4 \times 4$ | 512 (36) | 61.4 | 53.3 | 54.3 | 34.3 | 50.8 |

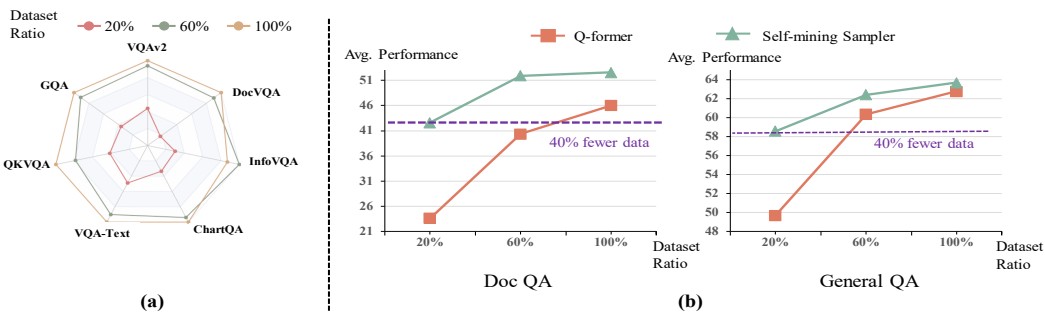

Figure 5: **(a) Ablation on data efficiency of HiRes-LLaVA.** We sample the training data mixture at ratios of 20%, 60%, and 100% and report the performance of our HiRes-LLaVAon seven benchmarks. **(b) Data efficiency comparison with Q-former and our proposed self-mining sampler (SMS).** The performance on 'Doc QA' is averaged from DocVQA, ChartQA and InfoVQA. The performance on 'General QA' is averaged from the other four benchmarks. Our SMS can use $40\%$ fewer data to achieve competitive performance compared with Q-former, indicating our method's efficiency. Note that both Q-former and our SMS apply one cross-attention block.

benchmarks and further improve all document-related benchmarks by increasing the base resolution and the number of vision tokens.

**Data efficiency analysis.** We evaluated the data efficiency of our method, HiRes-LLaVA, by subsampling the training data mixture at ratios of 20%, 60%, and 100%. Results in Fig. 5 (a) show that using the entire dataset achieves optimal performance. Remarkably, with only 60% of the data, performance remains above 90% of the full dataset's level, highlighting the potential for improved data efficiency. Additionally, we compared our self-mining sampler's efficiency against the commonly used Q-former in LVLMs. As depicted in Fig. 5 (b), our method performs competitively with Q-former even with only 20% of the data, demonstrating its effectiveness and efficiency.

## 5 CONCLUSION

In this paper, we present HiRes-LLaVA, a large visual-language model (LVLM) designed to efficiently address input fragmentation caused by current slicing-based high-resolution LVLMs. To evaluate this capability, we introduce a new benchmark, EntityGrid-QA, which includes identification, position, and counting tasks. Comprehensive experimental results on 11 popular existing benchmarks and EntityGrid-QA demonstrate the effectiveness of HiRes-LLaVA. Analytical evaluation and visualization results are provided for a deeper understanding of the model's performance.

**Limitations.** The samples in our constructed EntityGrid-QA are simple, lacking complex backgrounds, and the categories of entities and tasks are limited. In the future, we aim to create a more diverse dataset to better evaluate the performance of LVLMs in handling fragmented input.

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

## A  APPENDIX

### A.1  IMPLEMENTATION DETAILS

**Training Datasets.** Table 7 shows the detailed dataset construction of the capability enhancement stage of HiRes-LLaVA. Specifically, it has 830K captioning including the ShareGPT4V (Chen et al., 2023b), ShareGPT4o (Laboratory, 2023) and ALLAVA (Chen et al., 2024). The are 821K OCR

data from SynthDoG (Kim et al., 2022) including English OCR data and Chinese OCR data, MMC-Alignment (Liu et al., 2023c), UReader (Ye et al., 2023), K12 printed (100TAL, 2023) which is a short OCR dataset. There is also 200K text instruction data from Magpie Pro (Xu et al., 2024b), sampling from the generated data by llama3.1-70b, Llama3-70b and Qwen2-72B.

Table 8 shows the detailed construction of the 3M instruction tuning dataset. First, we remove 23K caption data and ShareGPT data from original LLaVA-158K (Liu et al., 2023e) and include GPT4V/GPT4o-generated caption data, i.e., LAION-GPT4v (LAION, 2023), ShareGPT4V (Chen et al., 2023b) and ShareGPT4o (Laboratory, 2023). We use ALLAVA instruction data (Chen et al., 2024). To enhance the common knowledge of our model, we convert the visual spatial reasoning (Liu et al., 2023a), AI2D (Kembhavi et al., 2016), and Science QA (Lu et al., 2022) training set into the instruct-tuning data. To activate the understanding science, we collect data from ViQuAE (Lerner et al., 2022) TextbookQA (Kembhavi et al., 2017), IconQA (Lu et al., 2021) and sampled 50k data from the Cambrian's Data Engine (Tong et al., 2024). We also collect document-oriented data from diverse datasets, includes ChartQA (Masry et al., 2022), DVQA (Kafle et al., 2018), PlotQA (Methani et al., 2020), OCRVQA (Mishra et al., 2019), ST-VQA (Biten et al., 2019), DocVQA (Clark & Gardner, 2018), InfoVQA (Mathew et al., 2022), DeepForm (Svetlichnaya, 2020), TAT-DQA (Zhu et al., 2022), TableFact (Chen et al., 2019), LRV-Chart(Liu et al., 2023b) and WebSRC (Chen et al., 2021). We merge some datasets from Cauldron (Laurençon et al., 2024), including RAVEN, ROBUT-SQA, ROBUT-WTQ, HiTab, IAM, Rendered Text, ORAND-CAR-A, Visual7W, Chart2Text, ai2d, vistext, Diagram-image-to-text.

**Module Design Details.** The self-mining sampler consists of one cross-attention block with an output layer norm. The cross-attention block has a cross-attention layer and a FFN. Both of them apply the residual shortcut. The cross-attention layer has two layer norm for the query and key/value, respectively. As for the SliceRestore Adapter, the parameters of the self-attention layer with the layer norm are initialized from the pretrained CLIP self-attention at the same depth. To provide the positional information between slices, we apply a 2D RoPE (Su et al., 2024; Sun et al., 2023) on the global fusion module.

**Evaluation Details.** We utilize the open-source evaluation tools, lmms-eval (Li* et al., 2024), to align our evaluation method to LLaVA-Next (Liu et al., 2024a).

Table 7: Datasets in the capability enhancement stage.

| Task | Datasets(# Sample) | Sum |
|---|---|---|
| Caption | ShareGPT4V(89k), ALLAVA4V(684k), ShareGPT-4O(57k). | 830K(44.8%) |
| OCR | SynthDoG-EN(300k), MMC-Alignment(200k), UReader(101k), K12 printed(120k), SynthDoG-ZH(100k). | 821k(44.4%) |
| Text | Magpie Pro(200k) | 200k(10.8%) |
| **Total** | | **1.8M** |

## A.2 PERFORMANCE COMPARISON OF THE SAME DATASET.

To demonstrate the effectiveness of our method, we compare the performance of LLaVA-1.5 and our method trained on the same data. Specifically, we train the both methods on two different scale training data set, i.e., LLaVA-655K (Li et al., 2023b) and LLaVA-655K with additional Doc-79K data. Results from Table 9 show that our method outperforms the LLaVA-1.5 under both training data sets, confirms that the superior performance can be attributed to the method itself rather than the volume of data.

Table 8: Summary of datasets used in the instruction tuning stage.

| Task | Datasets(# Sample) | Sum |
|------|--------------------|-----|
| **General QA** | LLaVA(135K), ALLaVA(660K) VQAv2(83K), GQA(72K), OKVQA(9K), A-OKVQA(66K), VSR(12K), ShareGPT4V(89K), TextCaps(22K), Laion-GPT4V(11K), ShareGPT-4O(57K), RAVEN(3K), Visual7w(14K), RefCOCO(48K), VG(86K) | 1.4M (48.0%) |
| **Science** | ScienceQA(19K), ai2d(14K), ViQuAE(4K), TextbookQA(21K), IconQA(30K), Data Engine(50K) | 139K(4.6%) |
| **Doc QA /OCR** | OCRVQA(80K), TextVQA(57K), SynthDog(30K), LLaVAR(39K), WikiTableQuestions(29K), KleisterCharity(15K), iiit(6K), MLHME(30K), VisualMRC(19K), ChartQA(48K), DocVQA(102K), InfoVQA(33K), DVQA(200K), PlotQA(10K), TAT-DQA(2K), TableFact(65K), WebSRC(5K) DeepForm(8K), Chart2text(27K) Vistext(10K), chrome writting(9K), IAM(6K), Rendered text (10K), Orand-CAR-A(2K), lrv-chart(2K), ROBUT-SQA(9K), ROBUT-WTQ(4K), Hitab(3K), Diagram-image-to-text(0.3K). | 0.9M(30.1%) |
| **Code Generation** | WebSight(50K) | 50K(1.7%) |
| **Text-only** | Magpie-Pro(150K), Evol(142K), mathinstruct(81K), mathplus(95K). | 469K(15.6%) |
| **Total** | | **3M** |

Table 9: Ablation study of different training data. Using the same training data, our HiRes-LLaVA consistently outperforms LLaVA-1.5, demonstrating the superior effectiveness of our approach.

| Model | Data | VQA-Text | ChartQA | DocQA | InfoVQA | Avg. |
|-------|------|----------|---------|-------|---------|------|
| LLaVA-1.5 | LLaVA-665k | 53.3 | 13.7 | 14.2 | 19.4 | 25.1 |
| LLaVA-1.5 | LLaVA-665k + Doc-79k | 53.3 | 23.8 | 22.6 | 26.0 | 31.4 |
| HiRes-LLaVA | LLaVA-665k | 62.4 | 19.8 | 37.7 | 26.0 | 36.4 |
| HiRes-LLaVA | LLaVA-665k + Doc-79k | 62.3 | 57.6 | 58.5 | 39.2 | 54.4 |

## A.3 EFFICIENCY ANALYSIS

**Comparison with other LVLMs.** To validate the efficiency of our method, we compare the computational cost, training, and inference times with various LVLMs in Table 10. For computational cost, we report the FLOPs of the ViT backbone, connector, and LLM components for each model. Experimental results demonstrate that HiRes-LLaVA, despite processing inputs at twice the resolution of LLavA-Next ($1344^2$ vs. $672^2$), is able to reduce training time by approximately 74%.

**Comparison with other downsampling methods.** We also compare the FLOPs and training time of our proposed downsampling strategy SMS with other vision token downsamplers, including ConcatChannel (Jun Chen & Elhoseiny, 2023), Q-Former (Bai et al., 2023a), and C-Abstractor (Cha et al., 2023), as shown in Table 11. The results show that our SMS, even when combined with additional components like SRA, achieves competitive efficiency compared to existing state-of-the-art downsamplers.

Table 10: Comparison of the efficiency of different models. Note that training time is assessed under the SFT setting on a machine with 8 V100 GPUs. The inference time is assessed on the InfoVQA benchmark with 6096 images by using the lmms-eval. Note that using the same batch size per device and resolution, LLaVA-Next would be out of the memory. The ratios of training time for ours relative to LLaVA-Next are marked in **purple**.

| Training batch size | Inference Resolution | *FLOPs* | | | Training time | Inference time |
|---|---|---|---|---|---|---|
| | | ViT | Connector | LLM | | |
| *HiRes-LLaVA* | | | | | | |
| 2 | 1344x1344 | 6.6 T | 195.2 G | 37.1 T | 60.7h **(15.9%)** | 15.4m |
| *HiRes-LLaVA w/o SRA* | | | | | | |
| 2 | 1344x1344 | 6.5 T | 195.2 G | 37.1 T | 59.5h **(15.6%)** | 12.9m |
| *LLaVA-Next (LLaVA-1.6)* | | | | | | |
| 2 | 1344x1344 | | Out of the memeory | | | |
| 1 | 672x672 | 1.9 T | 120.8 G | 44.0 T | 381.0h | 13.2m |

Table 11: Ablation study of the efficiency of individual components for different downsamplers. We assume the inputs are an image with 16 slices and 100 text tokens. Note that no downsampling method causes out-of-memory (OOM) issues during training. Training time is assessed under the SFT setting on a machine with 8 V100 GPUs.

| *Components* | | *FLOPs* | | | Training Time |
|---|---|---|---|---|---|
| Downsampler | SRA | ViT | Sampler | LLM | |
| NoDownsample | ✗ | 6.5 T | 410.8 G | 148.3T | - |
| ConcatChannel | ✗ | 6.5 T | 164.3 G | 37.1 T | 58.6h |
| Q-Former | ✗ | 6.5 T | 205.5 G | 37.1 T | 58.9h |
| C-Abstractor | ✗ | 6.5 T | 258.2 G | 37.1 T | 60.7h |
| SMS | ✗ | 6.5 T | 195.2 G | 37.1 T | 59.5h |
| SMS | ✓ | 6.6 T | 195.2 G | 37.1 T | 60.7h |

### A.4 MORE VISUALIZATION

**Samples from EntityGrid-QA Benchmark.** We illustrate three examples from our proposed EntityGrid-QA benchmark in Fig. 6. These three samples visualize examples of the three tasks in the benchmark we proposed. For each task, we write or paste the digital number or object directly onto each position of an empty image, and ask questions to the models.

**More Qualitative Results.** To further validate the effectiveness of our model, we illustrate the more qualitative results of InfoVQA, ChartQA and V* Benchmark in Fig. 7 and Fig. 8. Moreover, we give two qualitative examples to present the HiRes-LLaVA's capability of generating HTML code when given a website image in Fig. 9.

### A.5 BROADER IMPACTS

The development of HiRes-LLaVAadvances the field of vision-language models and has broad implications for various applications, including document analysis, medical imaging and remote sensing. However, alongside these potential benefits, there are considerable concerns.

HiRes-LLaVA, not having undergone rigorous safety training, might generate harmful or inappropriate content, leading to legal and ethical issues. Furthermore, its enhanced ability to process high-resolution inputs could be misused for creating misleading news, contributing to disinformation. These potential negative impacts highlight the need for careful management and ethical guidelines in the deployment of such technologies.

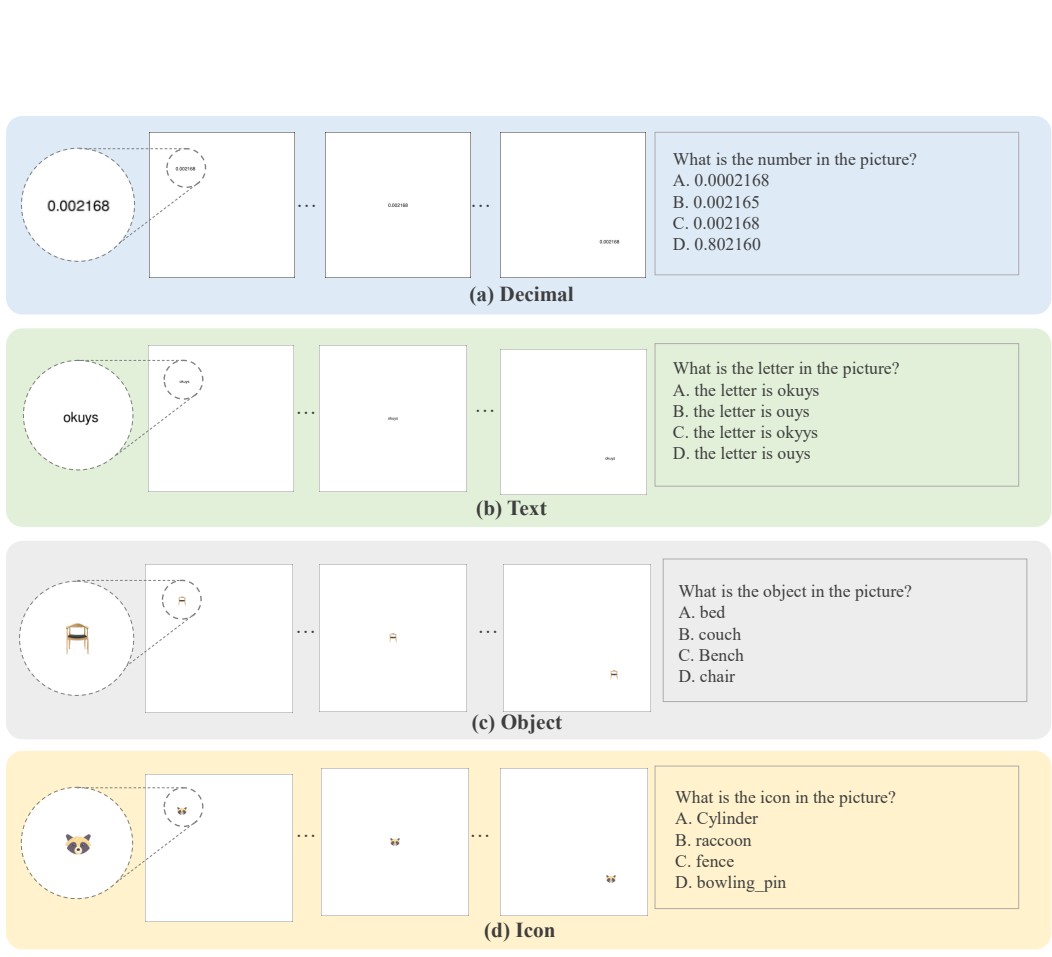

Figure 6: **Examples of our proposed EntityGrid-QA Benchmark.**

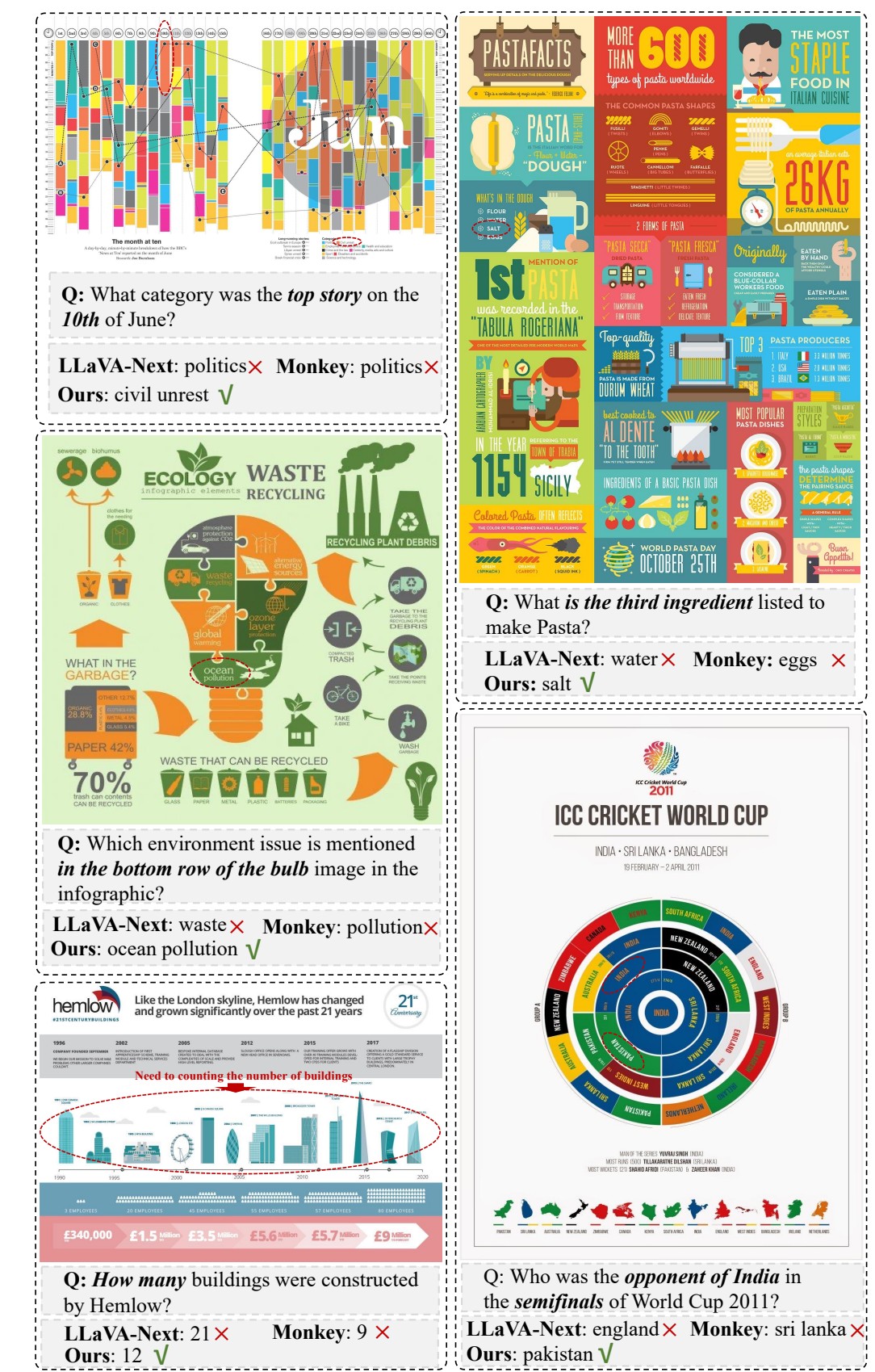

**Q:** What category was the *top story* on the *10th* of June?

**LLaVA-Next**: politics ✗ **Monkey**: politics ✗
**Ours**: civil unrest ✓

**Q:** Which environment issue is mentioned *in the bottom row of the bulb* image in the infographic?

**LLaVA-Next**: waste ✗ **Monkey**: pollution ✗
**Ours**: ocean pollution ✓

**Q:** *How many* buildings were constructed by Hemlow?

**LLaVA-Next**: 21 ✗ **Monkey**: 9 ✗
**Ours**: 12 ✓

**Q:** What *is the third ingredient* listed to make Pasta?

**LLaVA-Next**: water ✗ **Monkey**: eggs ✗
**Ours**: salt ✓

**Q:** Who was the *opponent of India* in the *semifinals* of World Cup 2011?

**LLaVA-Next**: england ✗ **Monkey**: sri lanka ✗
**Ours**: pakistan ✓

Figure 7: **Qualitative results from InfoVQA (Mathew et al., 2022).**

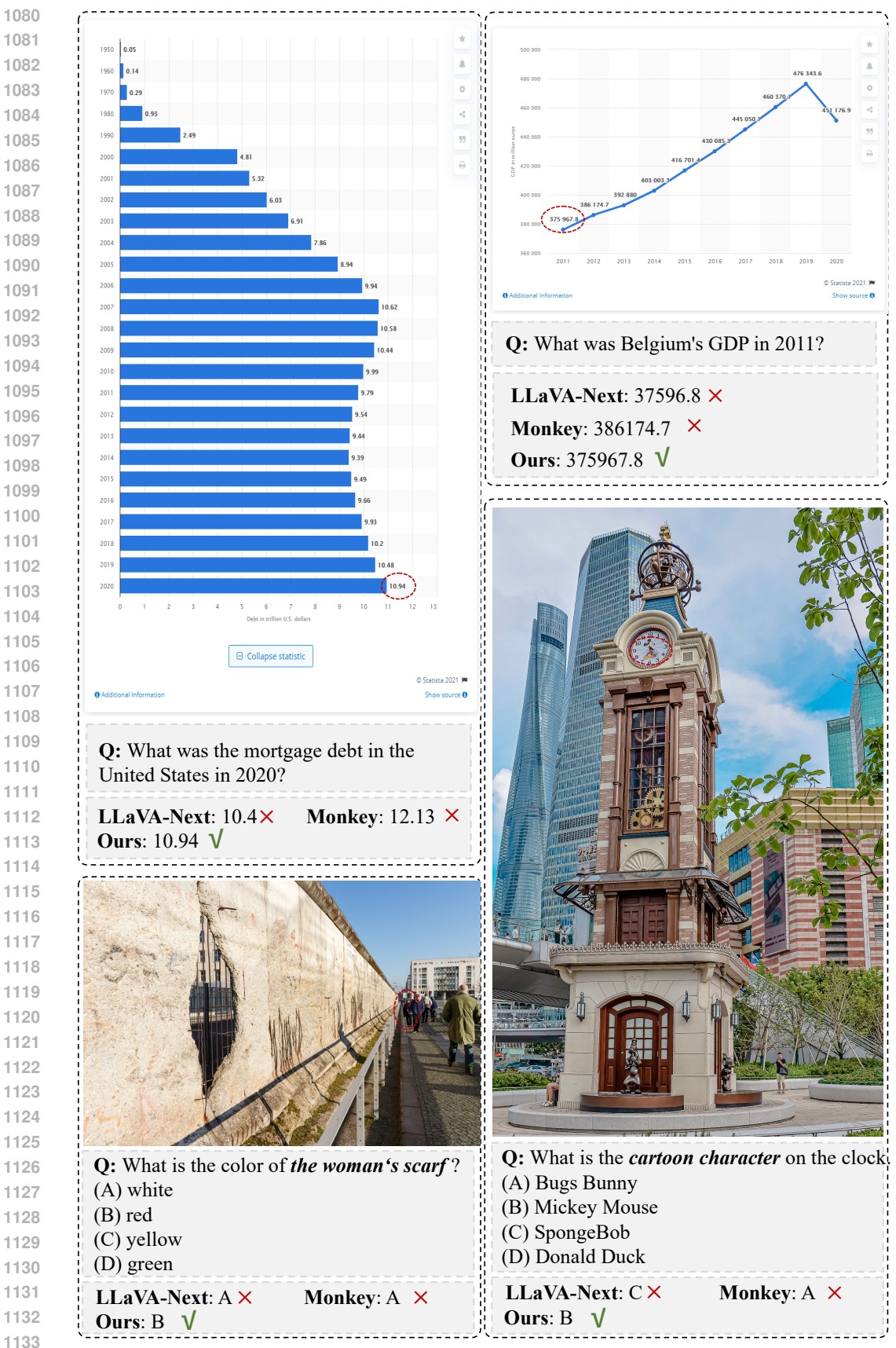

Figure 8: **Qualitative results from ChartQA (Masry et al., 2022) and Vstar Benchmark (Wu & Xie, 2023).** We use the red circle to highlight the answer target in the image.

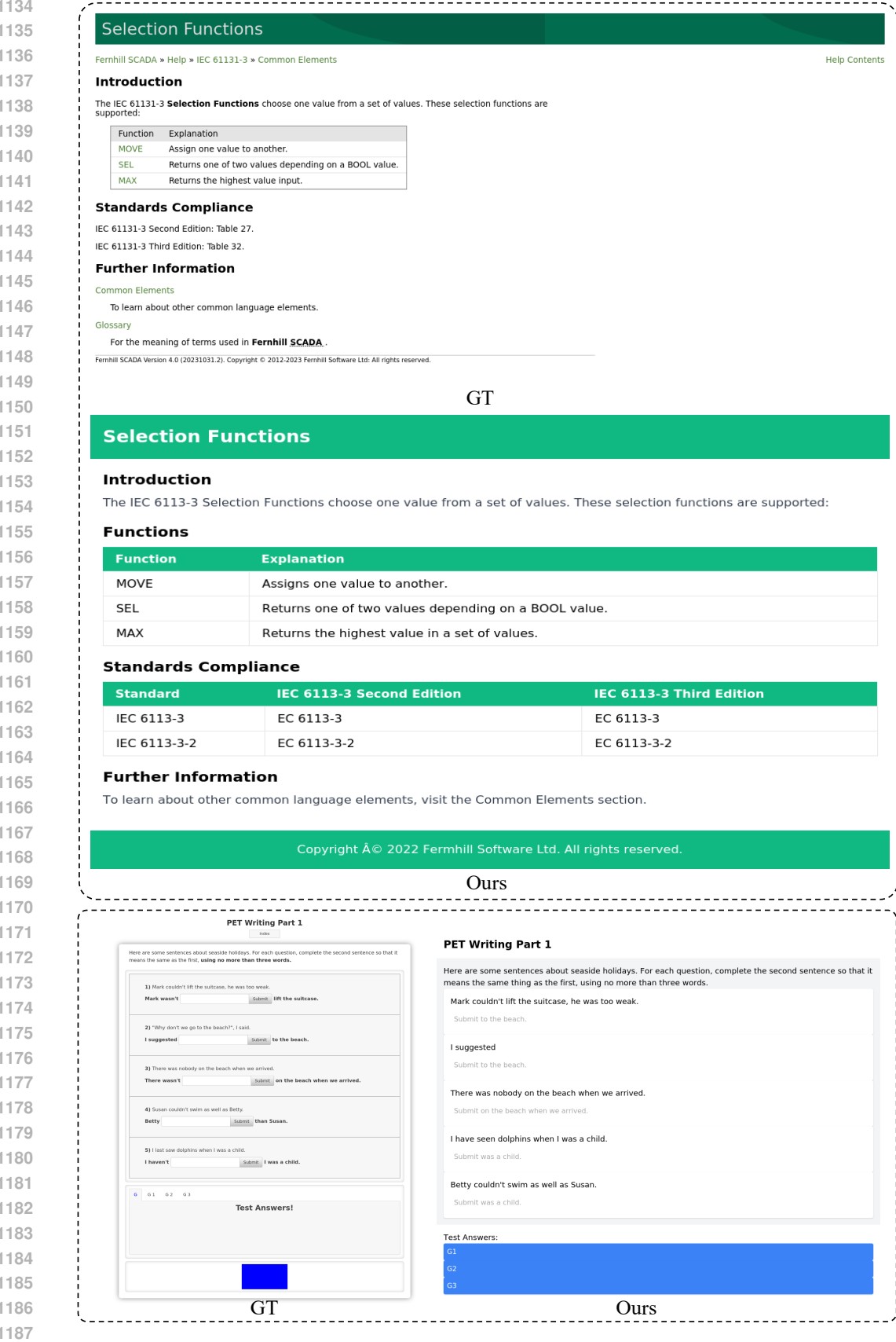

Figure 9: **Qualitative results on Image2HTML task (Si et al., 2024).** We visualize convert the generated html code to website image and compare to the input image.

