# OpenReview forum: "HiRes-LLaVA: Restoring Fragmentation Input in High-Resolution Large Vision-Language Models"
_ICLR.cc/2025/Conference — ICLR 2025 Conference Withdrawn Submission_

### Official Review · Reviewer_xEnX · 2024-11-03

**Soundness:** 3
**Presentation:** 3
**Contribution:** 3
**Rating:** 6
**Confidence:** 4

**Summary:**

This paper proposes the framework HiRes-LLaVa to address the issues of high training cost and poor image fragment continuity in LVLM under high-resolution image inputs. HiRes-LLaVa supports arbitrary resolution inputs while maintaining the original image's contextual and geometric continuity. HiRes-LLaVa incorporates the SlicerRestore adapter (SRA) in the vision encoder to extract global and local information, reconstructing image fragment features into the original size. To reduce the computational cost, the paper also introduces the image token compression method Self-Mining Sampler (SMS) to replace the Q-former. Across 9 image-text benchmarks, HiRes-LLaVa supports higher image resolutions and consistently outperforms or compares to baseline models. This work also introduces a new benchmark, Entity-QA, to evaluate the model's ability to handle fragmented inputs. On Entity-QA, HiRes-LLaVa demonstrates superior fragment handling capabilities compared to current state-of-the-art models. The paper conducts thorough ablation experiments to validate the proposed new modules, showing that the method not only improves the performance of models under different vision encoders or llms, but also achieves higher data efficiency and faster training convergence compared to existing models.

**Strengths:**

1. Theoretically, The method proposed in this paper can handle outputs of any resolution and shows a fast convergence in training and significant performance improvement, outperforming the baseline on 8 benchmark results.
2. A new and simple benchmark to evaluate the model for handling fragmented inputs is proposed, demonstrating the effectiveness of the SRA module in this paper.
3. The ablation experiments are comprehensive, showing the effectiveness of the proposed method across multiple models, with sustained gains as the resolution gradually increases.

**Weaknesses:**

1. The experimental setup description is unclear. Section 4.1 mentions a 3-stage training process, but section 4.3 gives out a LoRA finetuning is applied, and also claims the model is trained from scratch in comparison with Monkey.

**Questions:**

1. In section 4.3, it is mentioned that this work uses the LoRA fine-tuning. At which stage of training is this applied? Is HiRes-LLaVa trained on a pretrained model or from scratch?

---

### Official Review · Reviewer_cTLd · 2024-11-04

**Soundness:** 2
**Presentation:** 3
**Contribution:** 2
**Rating:** 3
**Confidence:** 4

**Summary:**

This paper introduces HiRes-LLaVA, aiming at restoring fragmentation input for high resolution VLMs. Two important components are proposed: the SliceRestore adapter to fuse the global and local features, and the self-mining sampler to better initialize the query and narrow the receptive field that per query needs to compress for faster congerence and preserving the positional information. Additionally, a new benchmark, EntityGrid-QA, is created to evaluate the model's ability to address input fragmentation. The proposed model outperformed prior works on various benchmarks including general benchmarks as well as the proposed EntityGrid.

**Strengths:**

1. The proposed self-mining sampler demonstrates higher data efficiency compared to Q-former and enhances the model's performance.

2. The newly introduced EntityGrid-QA serves as a valuable synthetic benchmark, offering a more comprehensive evaluation of MLLMs in handling information at the fragmentation boundary.

3. The model has a sota performance on various tasks including doc related tasks and science related tasks.

**Weaknesses:**

1. My primary concern lies with the contribution of the SliceRestore adapter (SRA), which is intended to address the fragmentation issue. Based on the ablation study, it appears that the main performance improvement stems from the self-mining sampler (SMS) rather than the SRA. This raises uncertainty about the actual utility of the SRA; if it proves ineffective, the fragmentation issue that this paper aims to address remains unresolved.

2. My second concern is the fairness of the comparison between the proposed model and the baselines. In Table 1, the proposed model was trained on different data from that of the other models, making it challenging to draw conclusions about whether the architecture itself outperforms others. While Table 9 attempts a comparison using a fixed dataset and contrasts the proposed model with LLaVA-1.5, this remains problematic as LLaVA-1.5 is not a high-resolution model. It is unclear if the performance gains are due to higher resolution or the proposed SMS and SRA modules. A more appropriate comparison would involve a baseline model that uses the same number of input slices (i.e., matching the resolution of the proposed model) without SMS and SRA, and trained on the same dataset.

3. The constructed EntityGrid-QA benchmark may be too simplistic, and I suspect that a model utilizing an overlapping sliding window for input slices could easily circumvent the challenges this benchmark is designed to test.

**Questions:**

1. Does the SliceRestore adapter (SRA) meaningfully contribute to solving the fragmentation issue, or is the performance gain primarily due to the self-mining sampler (SMS)? From my perspective, SMS is not designed to solve the fragmentation issue but to be provided as a better substitution for Q-former.

2. Could the performance gains in Table 9 be due to higher resolution rather than the SMS and SRA modules?

3. Is the EntityGrid-QA benchmark too simplistic, allowing models with an overlapping sliding window to bypass its intended challenges?

---

### Official Review · Reviewer_wYnJ · 2024-11-04

**Soundness:** 3
**Presentation:** 3
**Contribution:** 3
**Rating:** 5
**Confidence:** 4

**Summary:**

The paper presents HiRes-LLaVA, a framework designed to efficiently handle high-resolution image inputs in Large Vision-Language Models while maintaining contextual and geometric integrity. High-resolution inputs improve visual detail comprehension but increase computational costs and disrupt context when sliced into patches. HiRes-LLaVA introduces the SliceRestore Adapter (SRA) for reconstructing images from patches and the Self-Mining Sampler (SMS) for compressing vision tokens without losing context. The EntityGrid-QA benchmark is developed to test handling of context fragmentation, particularly in edge-related tasks. Experimental results demonstrate HiRes-LLaVA's superior performance, with a ∼9% improvement over existing LVLMs and enhanced efficiency in visual token downsampling using SMS.

**Strengths:**

1. The idea sounds and the paper is easy to follow.

2. Figures 2 and 3 are helpful for understanding.

3. Resolving fragment problem in the current dynamic slicing strategy for high-resolution input is important and valuable.

**Weaknesses:**

1. Table 3 does not provide the performance metrics for the SRA alone, SMS + SRA (L)

2. The main experiments is based on LLaVA-224, which is kind of outdate. It may be better and convicine to use LLaVA-1.5-336 as the baseline.

3. No computation cost and parameter flops analysis. It may be more convicine to provide a comparison between inference time, accuracy and number of tokens on different methods.

4. Please refer to my questions.

**Questions:**

* Table 5 ablates the SMS module, Token number vs performance. Could you also compare to other downsampler modules in Table 1, compare their token number and performance?

* Table 2 shows LLaVA baseline achieves best Discrepancy 1 and 2, it worth a explanation about why.

* When compare with other methonds on your EntityGrid-QA benchmark, do you fine-tune these models on your EntityGrid-QA datasets? How about the performance of HIRES-LLAVA without EntityGrid-QA trainingset?

* Compare Table 1 and Table 4 third row, we can see dataset has a very large gains on the performance. How about the performance of training LLaVA baseline, LLaVA-1.5-336 and LLaVA-Next on the same dataset as yours?

* You compare 4KHD in Table 2, but why don't you compare 4KHD in the Table 1?

---

### Official Review · Reviewer_wK3D · 2024-11-05

**Soundness:** 3
**Presentation:** 3
**Contribution:** 2
**Rating:** 5
**Confidence:** 5

**Summary:**

This work aims to improve the vision encoding module of Large Vision-Language Models for high-resolution images. A naive method is to slice the input image into uniform patches using sliding windows, and then encode each patch using the vision encoder. But this method has negative impacts on the cross-patch regions. This paper propose a new module, SliceRestore adapter, to introduce cross-patch interaction for joint feature extraction of multiple patches. The Self-Mining Sampler introduces better initializations for object query in Q-Former, which is easier for training. A new benchmark is also introduced to evaluate the model's ability for cross-patch region understanding.

**Strengths:**

1. The problem of high-resolution images encoding is very important for Large Vision-Language Models. A new method and benchmark are proposed to improve the accuracy and enrich the evaluation datasets, which would be interesting to the LVLM community
2. The two new modules, SliceRestore adapter and Self-Mining Sampler, are both well-motivated, and clearly-designed to improve the vision encoding of cross-patch regions, and improve the training efficiency for visual token compression.
3. The proposed method achieves strong performance not only on the newly-proposed benchmark, but also on many existing public datasets. The performance improvements are very impressive.

**Weaknesses:**

The major concerns is about the novelty of proposed modules, and insufficient literature review on similar approaches. (1) The key idea of SliceRestore adapter is to introduce the local and global fusion operations for multiple patches. However, this problem has been well studies in the vision-transformer paper before, like swin-transformer [1], PVT [2], Twins [3] and many others. The key of these work is also how to improve the communication of multiple small patch during multiple layer of feature extraction. A more through comparison should be provided. (2) The idea of using pooled image feature as object queries or learning object queries from local regions has been well studied in many DETR-style works before, Conditional DETR [3], DAB-DETR [4] and many others. The novelty of the proposed two modules is limited, considering the broader related literatures.

[1] Liu, Ze, Yutong Lin, Yue Cao, Han Hu, Yixuan Wei, Zheng Zhang, Stephen Lin, and Baining Guo. "Swin transformer: Hierarchical vision transformer using shifted windows." In Proceedings of the IEEE/CVF international conference on computer vision, pp. 10012-10022. 2021.

[2] Wang, Wenhai, Enze Xie, Xiang Li, Deng-Ping Fan, Kaitao Song, Ding Liang, Tong Lu, Ping Luo, and Ling Shao. "Pyramid vision transformer: A versatile backbone for dense prediction without convolutions." In Proceedings of the IEEE/CVF international conference on computer vision, pp. 568-578. 2021.

[3] Chu, Xiangxiang, Zhi Tian, Yuqing Wang, Bo Zhang, Haibing Ren, Xiaolin Wei, Huaxia Xia, and Chunhua Shen. "Twins: Revisiting the design of spatial attention in vision transformers." Advances in neural information processing systems 34 (2021): 9355-9366.

[4] Meng, Depu, Xiaokang Chen, Zejia Fan, Gang Zeng, Houqiang Li, Yuhui Yuan, Lei Sun, and Jingdong Wang. "Conditional detr for fast training convergence." In Proceedings of the IEEE/CVF international conference on computer vision, pp. 3651-3660. 2021.

[5] Liu, Shilong, Feng Li, Hao Zhang, Xiao Yang, Xianbiao Qi, Hang Su, Jun Zhu, and Lei Zhang. "Dab-detr: Dynamic anchor boxes are better queries for detr." arXiv preprint arXiv:2201.12329 (2022).

**Questions:**

1. what is the stride of the depth-wise convolution in L211? In table 6, it looks that the stride equals the kernel size. But a more natural way is to set stride=1 regardless of the kernel size.
2. How many entities are selected in a generated image in L274-L282. It is unclear what would the model performs with only 1 object and multiple objects.
3. The training pipeline is very complicated with three stages, compared to LLaVA. Can the authors make it more clear of the training dataset, tunable parameters in each stage?

---

### Note · Authors · 2024-11-15

I have read and agree with the venue's withdrawal policy on behalf of myself and my co-authors.